# An In Silico Study Based on QSAR and Molecular Docking and Molecular Dynamics Simulation for the Discovery of Novel Potent Inhibitor against AChE

**DOI:** 10.3390/ph17070830

**Published:** 2024-06-25

**Authors:** Meriem Khedraoui, Oussama Abchir, Hassan Nour, Imane Yamari, Abdelkbir Errougui, Abdelouahid Samadi, Samir Chtita

**Affiliations:** 1Laboratory of Analytical and Molecular Chemistry, Faculty of Sciences Ben M’Sik, Hassan II University of Casablanca, Casablanca 20670, Morocco; meriemkhedraoui5@gmail.com (M.K.); oussamaabchir12@gmail.com (O.A.); hassannour737@gmail.com (H.N.); yamariimane86@gmail.com (I.Y.); a_errougui@yahoo.fr (A.E.); 2Department of Chemistry, College of Science, United Arab Emirates University, Al Ain P.O. Box 15551, United Arab Emirates

**Keywords:** Alzheimer’s disease, acetylcholinesterase, inhibitors, QSAR, drug design, MLR, ADMET, molecular docking, molecular dynamics

## Abstract

Acetylcholinesterase (AChE) is one of the main drug targets for treating Alzheimer’s disease. This current study relies on multiple molecular modeling approaches to develop new potent inhibitors of AChE. We explored a 2D QSAR study using the statistical method of multiple linear regression based on a set of substituted 5-phenyl-1,3,4-oxadiazole and N-benzylpiperidine analogs, which were recently synthesized and proved their inhibitory activities against acetylcholinesterase (AChE). The molecular descriptors, polar surface area, dipole moment, and molecular weight are the key structural properties governing AChE inhibition activity. The MLR model was selected based on its statistical parameters: R^2^ = 0.701, R^2^test = 0.76, Q^2^CV = 0.638, and RMSE = 0.336, demonstrating its predictive reliability. Randomization tests, VIF tests, and applicability domain tests were adopted to verify the model’s robustness. As a result, 11 new molecules were designed with higher anti-Alzheimer’s activities than the model molecule. We demonstrated their improved pharmacokinetic properties through an in silico ADMET study. A molecular docking study was conducted to explore their AChE inhibition mechanisms and binding affinities in the active site. The binding scores of compounds M1, M2, and M6 were (−12.6 kcal/mol), (−13 kcal/mol), and (−12.4 kcal/mol), respectively, which are higher than the standard inhibitor Donepezil with a binding score of (−10.8 kcal/mol). Molecular dynamics simulations over 100 ns were used to validate the molecular docking results, indicating that compounds M1 and M2 remain stable in the active site, confirming their potential as promising anti-AChE inhibitors.

## 1. Introduction

Alzheimer’s disease (AD) is a neurodegenerative disease that causes progressive cognitive decline [1] and is the most common cause of dementia worldwide [2]. AD is a fatal disease [3] and is classified as the fourth most common cause of death in developed countries [4]. AD is not limited to memory impairment and has other effects, such as loss of attention, and behavioral and linguistic abilities [5], resulting in the patient no longer being able to perform activities of daily living. People aged 65 years or older are at the greatest risk of developing AD [6], and the number of patients will increase as the population ages [7]. An estimated 55 million people suffer from dementia worldwide in 2020 [8], and the number of people with dementia is expected to reach 131.5 million by 2050 [9,10]; therefore, AD poses challenges to public health [11], which is why it is considered a global health problem [12].

At present, the cause of AD is not yet understood and remains unclear [13]. When discovered in 1906 by Alois Alzheimer when he examined the brain of a deceased woman who suffered from memory loss, he noticed changes in the brain tissue, including the appearance of abnormal clumps called senile plaques and tangles, which remain the main features of AD. At present, many hypotheses have been proposed to explain AD, such as the Aβ deposition hypothesis, the tau protein hypothesis, and the cholinergic hypothesis; however, they do not present the exact cause of AD [14]. The cholinergic hypothesis is one of the most studied major theories. This theory holds that AD patients suffer from a decrease in cholinergic neurons [15], which leads to a decrease in ACh. ACh is a neurotransmitter through which communication occurs between neurons in the brain, and which is involved in memory, learning, and other behavior; moreover, deficiency of this neurotransmitter ACh leads to cognitive disorders [16]. Based on these facts, the therapeutic approach currently considered for AD consists of increasing cholinergic neurotransmission in the brain by reducing unwanted hydrolysis of ACh by AChE. Acetylcholinesterase is an enzyme involved in stopping impulse transmission through the rapid hydrolysis of ACh to acetic acid and choline [17]. Therefore, acetylcholinesterase inhibitors are administered to block AChE activity to stop the breakdown of acetylcholine and maintain its levels in the brain. The three AChE inhibitors currently used in the treatment of AD, namely, Donepezil, Rivastigmine, and Galantamine, provide only symptomatic relief with limited therapeutic effect [18]. Thus, research into new potent AChE inhibitors is essential.

To design new AChEI inhibitors quickly, the use of mathematical models that relate molecular structures to their biological activity is essential. This is a Quantitative Structure–Activity Relationship (QSAR) method based on the principle that the molecular structure of chemical compounds is correlated with biological activity [19]. This approach is more commonly used in medicinal chemistry, where drug design becomes more rational with a minimum of costly and time-consuming experiments. Much research has been carried out in the field of AD, using the QSAR tool to explore new synthesizable drugs. Hassan et al. applied the QSAR constructed using multiple linear regression analysis on a set of BuChE inhibitors consisting of 36 carbamate derivatives. Based on the model developed in their study, they were able to design new inhibitors with higher anti-BuChE activities [20]. Bala et al. developed a QSAR model based on the use of chalcone derivatives as acetylcholinesterase inhibitors, and, after validating its predictive power through validation tests, they were able to predict the inhibitory activity of new anti-AChE compounds designed with chalcones fused with quinoxaline [21].

In this study, we plan to design new potential inhibitors of AChE using the QSAR tool, in addition to resorting to the use of other modeling techniques such as molecular docking and molecular dynamics.

## 2. Results and Discussion

The QSAR model is established by multiple linear regression and is represented by Equation (1) the model shows that pIC_50_ activity is expressed by three descriptors, namely polar surface area (PSA), dipole moment (µ), and molecular mass (MW).
pIC_50_ = 2.587 − 0.007 PSA + 0.163 µ + 0.010 MW(1)

Model acceptance was examined more closely by considering many statistical parameters such as squared correlation coefficient (R^2^), low root mean square error (RMSE), cross-validated correlation coefficient (Q^2^_CV_), and (R^2^_test_) for external validation, Fisher value (F test).

The coefficient of determination of the training set R^2^ = 0.64 is remarkably high, which makes it quite acceptable. As shown in Figure 1a, there is a correlation between the activities predicted by the model and the experimental. However, it should be noted again that R^2^ is only a measure of the goodness of fit between the values predicted by the model and the experimental values and does not reflect the predictive power of the model at all [22]. Therefore, parameters need to be further established to highlight the predictive ability of the models. The cross-validated Q^2^_CV_ was performed using the leave-one-out (LOO) method to measure the internal predictive power of the model. Since the true usefulness of a QSAR model lies in its ability to accurately predict the modeled properties of new chemicals that were not used in the development of the model, a realistic assessment of the true predictive power of the model must be made, and this is accomplished by measuring the external validation test indicator, which produces the statistical correlation between the actual activities of the compounds in the test set and those that were predicted. According to the acceptance thresholds given in Table 1 below, the generated model was considered to have predictive power as long as the correlation coefficient R^2^ of the training set is of value 0.64, of plus the cross-validated correlation coefficient Q^2^_CV_ = 0.56, and correlation coefficient test R^2^_test_ = 0.72 (as shown in Figure 1b), shows that the developed model has internal and external predictivity. The RMSE value of 0.35 and the value of the Fischer test F = 21.54, p < 0.0001 show that the generated model is statistically significant. The validation parameters generated in this study were within the acceptable threshold parameters, this model demonstrates their predictive power.

### 2.1. Randomization Test

The correlation of the developed model was also validated using the randomization test. Table 2 presents a comparison of the results of the parameters (R^2^, Q^2^) of this test with that of the original model. Indeed, the parameters of the random test do not correspond to those of the original model, which proves that the correlation between pIC_50_ and the three selected descriptors is not due to chance.

### 2.2. Variance Inflation Factor 

We applied the Variance inflation factor (VIF) test to examine multi-collinearity between descriptors, and the VIF scores presented in Table 3 are all between (1–2.5) and below the threshold (VIF < 10) [25], indicating low intercorrelations between the descriptors and the absence of multi-collinearity between the three descriptors [26].

### 2.3. Applicability Domain

Figure 2 elucidates the Williams plot for the 2D-QSAR model. This is a plot of the normalized residuals versus the lever values of the molecules, within which the model predictions are reliable [27,28]. Based on the results observed in the graph, no outliers outside of AD were detected. The compounds studied were limited respectively at the normal residual limit of ±3 and the leverage threshold of h* = 0.300, which means that the dataset was predicted correctly.

### 2.4. Design of New Inhibitors and Interpretation of Model Descriptors

The three molecular descriptors presented in the QSAR model provide a guide for the design of new derivatives with increased anti-Alzheimer activities biological activity, which are of great importance for improving activity. The relative significance of the descriptors presented in the model was determined on the basis of their standardized regression coefficients, as shown in Figure 3. The standardized regression coefficients for each descriptor provide important information about the effect of the molecular descriptors and their degree of contribution to the model developed. The order of contribution is as follows: MW > µ > PSA.

The PSA descriptor has a negative contribution to pIC_50_. pIC_50_ is inversely proportional to PSA which requires a reduction in this descriptor to increase activity.

The second parameter is the dipole moment descriptor, which is an electronic factor and is related to polar interactions [29] that are used as a measure of asymmetry in charge distribution, indicating the degree of charge separation in a molecule. The dipole moment has been shown to have a positive contribution and favorable impact on anti-AChE activity. We also took into account the positive impact of mass on pIC_50_; moreover, biological activity is positively influenced by the increase in molecular mass. We carried out molecular variation by selecting the molecule presenting the highest level of anti-Alzheimer activity; then, the molecule underwent structural modifications by substitution of atoms and atom groups on different sites. We, therefore, designed 11 molecules that are shown in Table 4, with their values of pIC_50_ predicted using the model of eq 1 and the calculation of its leverage values to identify the compounds belonging to the domain of the application of the aberrant one.

Among the 11 molecules designed, the compounds following M3, M4, M5, M7, M8 and M11, have inhibitory activities lower than the reference value that corresponds to (pIC_50_ = 7.259), and what concerns M1, M2, M6, M9, and M10 have activities higher than the reference one. Their leverage values were calculated and compared to the threshold value (h* = 0.450). Moreover, as presented in Table 4, the designed compounds were located in the AD space of the model.

### 2.5. In Silico Drug Resemblance and ADMET Pharmacokinetic Prediction

The lack of efficacy and safety of candidate drugs is one of the main reasons for the clinical attrition of drugs. Therefore, the concepts of drug similarity and pharmacokinetic parameters have been used to filter compounds with undesirable properties.

The Lipinski rule is the most famous rule of drug likeness, and it defines the acceptable limits of four molecular physicochemical properties for orally active compounds [30]. The drug likeness properties of our proposed compounds were predicted in silico, and the compounds will, therefore, be selected based on Lipinski’s rule of five criteria. As illustrated in Table 5. It is seen that the molecular weight of the compounds is less than 500 according to Lipinski’s rule. In contrast, compounds with higher molecular weight have poorer intestinal and blood–brain permeability [31], suggesting that the proposed compounds are capable of being orally active. On the other hand, the lipophilicity calculated using the partition coefficient between water and n-octanol [32] revealed values around 6, which was greater than 5 and, therefore, indicated a slightly poor absorption, as well as when the number of hydrogen bond acceptors and donor meets the criteria within the appropriate Lipinski interval (HBD < 5 and HBA < 10), respectively. Usually, the active compound administered orally should not exceed more than two violations of Lipinski’s rule. We note that the compounds proposed in this study did not violate the rule beyond the maximum permitted limits, which demonstrated their properties as medicinal products [33]. Pharmacokinetic parameters included absorption, distribution, metabolism, and excretion. We also assessed the toxicity of the proposed compounds, as many drug candidates may have low ADMET properties and be toxic. Therefore, ADMET properties are of great importance in drug discovery, as they mitigate potential issues during drug testing in clinical trials [34]. All the proposed compounds show a value greater than 80%, which indicates that the compounds are well absorbed in the human intestine. The prediction of Caco-2 permeability is often used to predict the absorption of drugs administered orally; therefore, the proposed compounds have a predicted value greater than 0.9, which indicates that they are all permeable to Caco-2 cells except the M4 molecule, which did not meet this condition. Drug discovery for central nervous system (CNS) diseases faces challenges, one of which is the obstacle of the sufficient penetration of drugs into the blood–brain barrier (BBB) [35]. Drugs intended for the treatment of CNS must necessarily have a high chance of penetrating the BBB. The BBB is filled with tightly bound endothelial cells, which limits the ability of the compound to be transported into the bloodstream via the administered route [36].

Substances with LogBB < −1 diffuse poorly in the brain, while compounds with LogBB > 0.3 have the ability to cross the BBB. In addition, substances with Log PS > −2 enter the CNS, and substances with Log PS < −3 would be more difficult to transport through the CNS [37]; in fact, compounds M1, M2, M6, and M9 show the ability to cross the BBB and are classified as active on the CNS, and the inactive compounds is compounds M10 which have values less than −1. In a metabolism analysis, the cytochrome P450 2D6 model predicts the inhibitory and non-inhibitory behavior of the 2D chemical structure. CYP2D6 is an important enzyme system for drug metabolism in the liver, and the inhibition of CYP2D6 contributes to amplifying the effect of various drugs, which can lead to dangerous levels [38]. It should be noted that the compounds proposed are not CYP2D6 inhibitors and therefore are not expected to affect hepatic dysfunction when administered [39]. Excretion is the process by which the body gets rid of waste/toxic products. Drug excretion may occur through the kidneys and/or liver, where drugs are eliminated as urine or bile, respectively. Complete drug clearance refers to the half-life of the drug: the lower its value, the longer the half-life of the compound [40]. All compounds have low clearance values, which means they have high half-lives. Furthermore, one of the conditions for successful drug development is the safety of the molecules; therefore, toxicity prediction is an important criterion in compound testing. As shown in Table 6, using four toxicological parameters—hepatotoxicity, mutagenicity, skin sensitization, and acute toxicity in rats (LD_50_ value)—the results indicate that the newly designed molecules are non-mutagenic, according to AMES test data. Hepatotoxicity and skin sensitization tests showed that none of the molecules were toxic except the M10 molecule, which is hepatotoxic, and the LD50 values indicate that the compounds are lethal only at very high doses.

These results indicate that only compounds M1, M2, and M9 presented an ADMET profile. They are, therefore, potentially interesting candidates for molecular docking studies.

### 2.6. Molecular Docking

We tested the efficiency of molecular docking algorithms to guarantee the accuracy of the results. As shown in Figure 4, the redocking ligand was superimposed on the original ligand in the active site, and the root mean square deviation (RMSD) is 0.28 Å, indicating the reliability of the docking protocol as it can reproduce the experimental results.

The proposed compounds M1, M2, and M6 were chosen to study their mode of binding in the AChE groove. Various interactions were observed in the assembly of the ligands with AChE. In Figure 5a, compound M1 binds to AChE by establishing hydrophobic interactions involving residues such as Trp86, Trp286, Tyr341, Tyr72, Trp286, Val294, Tyr337, Phe338, Tyr341, and Trp286. A hydrogen bond is formed by Val294, a halogen bond by Trp286, and a Pi-Lone Pair bond by Phe338. All these interactions help to intercalate the M1 ligand into the active site, forming a stable complex.

In Figure 5b, an analysis of the binding of the M6 ligand in the active site reveals that the ligand is held in the active pocket by hydrophobic interactions with residues Phe338, Tyr341, Trp286, Trp86, and His447 and hydrogen bonds with residues Tyr133, Gly121, Glu202. Halogen interactions with Trp86, Gly120, Ser125, and Tyr337 were also observed.

With regard to the M2 ligand, an analysis of its interactions with the target protein, as shown in Figure 5c, revealed that the M2 ligand reacts with AChE by engaging in hydrogenic bonds via residues Phe295, Trp286, Tyr124, as well as halogen interactions with Val294 residues and hydrophobic interactions with residues Tyr337, Phe338, Trp86, Trp286. Regarding the reference drug, as shown in Figure 5d, Donepezil binds to AChE, through interactions interacting with the binding site, notably, hydrophobic interactions involving residues TRP86, TRP286, and TYR72, and it was also observed that a hydrogen bond was produced between residue PHE295.

All of the many non-covalent connections that make up the complexes constitute very important partners in the binding mechanism.

Given that the ligands bind to the active site of the receptor with similar residue involvement to the standard inhibitor, we found that there are common residues between the reference drug and the proposed ligands, notably, TRP86 and TRP286. The ligands show potential similarity to Donepezil in the mechanism of association, indicating that they could have the same therapeutic effect as Donepezil and that they would be competitive inhibitors of Donepezil [41,42].

In terms of the binding affinity of the proposed compounds, with respect to AChE, score functions were calculated using the Autodock vina software version 1.5.7, which is presented in Table 7 and is an estimate of the affinity between the protein and the small organic molecule. A score, therefore, does not predict an activity but an affinity [43] and measures the strength of the bonds established during the molecular docking of the proposed compounds. By comparing the binding energy of the compounds with those of the reference drug Donepezil, the results revealed that all three ligands with lower binding energy exhibited good binding affinity for AChE compared to Donepezil.

The molecular docking study identified the key residues involved in ligand binding to AChE. Among these residues, TRP86 and TRP286 were identified as crucial for inhibitor binding to AChE.

The study of mutagenesis at AChE makes it possible to identify the amino acid residues essential for enzyme function and to confirm their possible involvement in inhibitor binding [44]. It is plausible that the binding of Donepezil and ligands to AChE depends solely on interaction with these residues.

Subsequent mutagenesis studies have revealed that mutations in certain residues can have a significant impact on ligand binding affinity. For example, the mutation of aspartic acid at position 3.32 to alanine (D3.32A) in the serotonin G-protein coupled receptor (GPCR) 5-HT2C abolishes detectable binding of the standard 5-HT2C radioligand. This strongly suggests that this residue is essential for substrate binding to the receptor [45]. Therefore, the study of mutagenesis is essential to identify key residues, such as TRP86 and TRP286 in order to design potent inhibitors against AChE, potentially useful in the treatment of AD.

### 2.7. Molecular Dynamic and MMGBSA Calculations

Molecular docking provides a static perspective of the ligand’s position in the protein’s active site. To address protein flexibility, molecular dynamics simulations have been conducted to elucidate the variability that may occur in the protein–ligand system and to model how interactions occur in the complex in the physiological environment. Several publications emphasize that a compound with a higher score does not necessarily mean it is a potent compound, as the compound may escape from the binding pocket. Therefore, MD simulations are used as a baseline control before drawing conclusions from docking results.

The MD simulations were performed to explore the dynamic behavior of the complex upon ligand binding to the AChE protein. In the present study, we examine the molecular dynamics simulation results of 100 ns duration to obtain more information with respect to the predicted conformation, which includes RMSD diagrams, RMSF, and an interaction diagram. RMSD is a parameter that provides information on the variation of atomic positions relative to the starting structure obtained from molecular docking studies, this allowed us to analyze the evolution of root mean square deviations.

The RMSD diagram of AChE and its complexes with AChE–M1, AChE–M2, and AChE–Donepezil is shown in Figure 6. The RMSD profile of the AChE structure showed some initial structural variation, and, after a certain period of simulation, it stabilized around 1.9 Å (Figure 6a). The RMSD value of the protein in a complex with ligand M2 started to rise at the start of the simulation; moreover beyond 25 ns, the RMSD value turned out to be stable at 1.50 Å, which means that the protein in the complex did not have significant structural changes and that, between 75 ns and 85 ns, the protein in the complex made slight structural changes, with an RMSD value of 1.75 Å, after which it returned to its stability until the end of the simulation. Furthermore, the RMSD value of the M2 ligand was 0.9 Å after 60 ns and the M2 ligand only made minor structural changes that did not influence its stability in the active pocket, indicating that the M2 ligand remains stable in the active site (Figure 6c).

Similarly, the AChE–M1 complex reached its structural deflection stability with an RMSD value of 1.85 Å after 65 ns of simulation time. The RMSD plot of the M1 ligand gradually increases, and, after 20 ns, the M1 fluctuations remain stable at around 1.8 Å, suggesting that the M ligand bound to the AChE cavity during the MD simulation (Figure 6b). The AChE–Donepezil protein remained stable during the simulation with an RMSD of 1.50 Å. Donepezil’s trajectory showed a slight variation between 50 and 70 ns, but its RMSD did not increase significantly, remaining at around 5.1 Å (Figure 6d).

Additionally, we examined the root mean square fluctuation (RMSF) of the residues of all complex systems and the apoprotein to elucidate the local changes of the residues. Figure 7a shows that the RMSF analysis of AChE shows some high fluctuations of about 3 Å, and the residue oscillation of the AChE–M1 (Figure 7b), AChE–M2 (Figure 7c), and AChE–Donepezil (Figure 7d) complexes during the 100 ns simulation approximately fluctuate less than that of AChE residues. The RMSF plot reveals that the residues of these complexes (AChE–M1, AChE–M2, and AChE–Donepezil) are less flexible and no structural alteration was shown; therefore, the AChE protein in the complex is sufficiently stable, and the conformation presents low flexibility, which confirms the strong attachment of the ligands M1, M2, and Donepezil to it.

The intermolecular interactions between the protein and the ligand were studied over a simulation time of 100 ns using a simulation interaction diagram. Protein–ligand interactions play a key role in the association of the ligand and the protein. Figure 8a presents two types of contacts between the ligand M2 and AChE hydrogen bonds. In hydrogen bonding, residue PHE295 is involved with a formation rate of 44%, while hydrophobic interactions are exhibited by five residues, namely, TYR341, PHE338, PHE297, TRP286, TRP86, with formation rates of 59%, 19%, 22%, 100%, 78%, respectively.

For compound M1, SER125 had a water-bridge formation rate of 42%. Residues TRP86, TRP286, TYR337, TYR341, HIS447, and TYR449 contributed 90%, 60%, 30%, 90%, 45%, and 35%, respectively to the formation of hydrophobic interactions in Figure 8b. These interactions support the stable binding of the M1 and M2 ligands to the AChE protein.

On the other hand, Figure 8c revealed that the main interactions between Donepezil and the AChE pocket site included hydrogen bonds, hydrophobic interactions, and water bridges. Residues involved in these interactions were TYR72, ASP74, TRP86, TYR124, ALA127, LEU130, TYR133, TRP286, and TYR341. Hydrophobic interactions were specifically established with residues TYR72, TRP86, LEU130, TRP286, and TYR341, with formation rates of 35%, 49%, 22%, 70%, and 82%, respectively. Hydrogen bonds were formed with residues ASP74 and TYR124, with formation rates of 18% and 30%, respectively. Water bridges were created by residues ALA127 and TYR133, with occupancy rates of 20% and 35%, respectively.

The molecular dynamics (MD) complexes were subjected to post-MD analysis, using MM-GBSA to estimate their binding free energies. The free binding energies of the two complexes ranged from −64.78 kcal/mol (M1) to −80.66 kcal/mol (M2). Compared with the reference drug, Donepezil has a binding free energy of −65.38 kcal/mol, comparable with that of the M1 ligand. The negative ΔG values indicate spontaneous and strongly favorable interactions between the ligands and the protein. This confirms that the interactions formed between the proposed ligands and the protein favor the formation of stable complexes. The results indicate that the M1 and M2 ligands are relatively stable, highlighting their potential potency and stable interaction with the target protein to inhibit AChE activity.

## 3. Materials and Methods

### 3.1. Collects the Database and Calculates Molecular Descriptors

The 2D QSAR model was generated using a database limited to a relatively small set of N-benzylpiperidine and 5-phenyl-1,3,4-oxadiazoles analogs, as shown in Table 8. The database was taken from the literature [46,47] and consists of two series of analogs (5-phenyl-1,3,4-oxadiazole and N-benzylpiperidine) that have experimentally demonstrated AChE inhibitory activity; then, all their IC_50_ values were converted into logarithmic form (logIC_50_). The molecular structures of the database were drawn in ChemDraw version 20.0 [48] and their structures were optimized by minimizing the energy. The electronic descriptors were calculated using the Gauss View program version 09 [49], adopting the approach of the hybrid density functional theory B3LYP combining the three Becke parameters and the Lee–Yang–Parr exchange correlation functional with basis groups 6–31G (d, p). Moreover, using Chem3d software version 20.0, we calculated different molecular descriptors, in particular, constitutional descriptors, physicochemical descriptors, and geometric descriptors [48], which were used as input variables in the establishment of a 2D QSAR model, because they quantitatively translate the information chemicals relative to molecular structures [50].

### 3.2. Analysis of Correlation Matrices

In total, 50 obtained descriptors were preprocessed, and redundant and uninformative descriptors as well as zero and constant descriptors were excluded. A correlation matrix analysis was performed using XLSTAT software version 2013 [51] to reduce the number of descriptors, such that those strongly related to each other and those with weak correlation with biological activity were removed, leaving the descriptors with a strong correlation with the pIC_50_ value [52].

### 3.3. Data Split and Model Develop

At the initial stage of QSAR modeling, the data set was randomly split into two parts, one formed by the training set with a rate of 80% of the database which represents the input base on which the model will be adjusted, and the other fraction contains the test set trained by a rate of 20% of the compound used to examine the external predictivity of the model. Then, the QSAR model is modeled using the MLR multiple linear regression statistical technique using XLSTAT software.

### 3.4. Model Validation

The developed QSAR model is the subject of a validation study to examine credibility. Leave-one-out (LOO) cross-validation is used to evaluate model fit based on new data predictions that have, in turn, been removed and not used in model building. Moreover, depending on the value of the parameter Q^2^_cv_, we measure the internal predictive capacity of the model. Then, an external validation carried was out on the test set. Additionally, a randomization test was applied to check for the additional statistical significance of the relationship between model descriptors and biological activity [53]. In the model construction matrix, the pIC_50_ values are mixed, keeping the descriptors identical to those of the non-randomized model matrix, and 100 new QSAR models are created. The results of the cRp2 parameter must be greater than 0.5 [22] in order to prove that the model that was developed in this study is not the result of chance. The VIF test was applied to detect multi-collinearity between model descriptors, where a VIF value between 0 and 10 indicates that the descriptors do not significantly affect the estimation of the regression coefficients [54].

### 3.5. Applicability Domain

The aim of using the QSAR model is to predict and determine the pIC50 for new AChE inhibitors. It is desirable to evaluate the accuracy of the compound predictions so that they can be used in any confidence; thus, the reliability of the 2D-QSAR prediction is only valid if the compound to be predicted is in the domain of applicability (DA), which falls within the validation procedures implemented at the level of the OECD [55], the domain of applicability being the response and chemical structure space in which the model predicts reliably. Moreover, via DA, we can distinguish reliable and unreliable predictions [56]. The leverage method was adopted in this study to define the applicability domain, also known as the Williams diagram, which was plotted in IBM SPSS Statistics 26. It is based on leverage (*h*i) calculated in Equation (2) and on standardized residual compounds (SDR).
hi = x^T^_i_ (X^T^X)^−1^ x_i_(2)
where X is the descriptor matrix of the training set and xi is the descriptor vector of the compound. The leverage threshold value is defined as h* = (3 × (K + 1))/n, where n is the amount of training and K is the number of descriptors. Compounds with hi greater than the critical leverage threshold value are chemically different from the training set and report as outliers [57].

### 3.6. ADMET In Silico Pharmacokinetic Prediction

Over the past decade, approximately 50% of late-stage drug development failures have been due to unacceptable ADMET properties [58]. Reducing the attrition rate requires optimizing ADMET characteristics and respecting drug similarity from the start of drug discovery. In this research, the pharmacokinetic properties were predicted using the pkcsm server [59]. Additionally, drug similarity parameters including a logP value of ≤5, molecular mass (MW) of ≤500 Da, a number of hydrogen bond donors ≤5, and a number of hydrogen bond acceptors (N atoms and O) ≤10 [60] are simulated using the SwissADME server [61].

### 3.7. Molecular Docking

Molecular docking is one of the commonly used methods in computational chemistry to expedite drug discovery at an early stage [62]. We used them in this research to explore potential interactions between novel compounds designed as inhibitors and the binding site of the target protein of AD. The structure of the AChE target protein was uploaded on the Protein Data Bank (PDB) website “http://www.rcsb.org/pdb (accessed on 16 June 2024)” in a 3D form identified by the PDB code = 4ey7 and discovered through X-ray diffraction with a resolution of 2.35 Å. We then proceeded to prepare the protein in the Autodock program version 1.5.7 [63].

The protein preparation protocol was carried out as follows: removal of cofactors bound to the protein, the addition of hydrogen atoms along with the addition of Kollman charge and Gasteiger charges, and the insertion of missing atoms into residues incomplete. Finally, the prepared structure was saved as a pdbqt file.

Citing the active site through the position of the co-ligand at x = −14.108, y = −43.833, and z = 27.670, on which the grid is centered with dimensions of 40 Å × 40 Å × 40 Å and spacing of 0.375 Å, after docking between the ligand, the catalytic site is then carried out and a score function is calculated using the AutoDock vina software version 1.5.7 [64]. Furthermore, we used Discovery Studio software version 2017 for visualizing interactions between receptor residues and the ligand [65].

### 3.8. Molecular Dynamics Simulation

The complex structure was prepared for molecular dynamics simulation using Desmond (New York, NY, USA) [66]. The molecular dynamics simulations were performed for over 100 ns to study their stability.

In performing the solvation of the systems, we used the TIP3P (3-point transferable intermolecular potential) water model in an orthorhombic box, followed by neutralization with the addition of Na+, Cl- ions that are distributed randomly in the solvent system. The salt concentration was maintained at 0.15 M. After building the system, we used the OPLS3e force field constants [67] from the Desmond package to minimize the system framework. The system was then equilibrated for the duration of 1 ns of simulation via the NVT set and then equilibrated for 100 ns via the NPT set under a constant temperature of 300 K and a pressure of 1 ATM (101,325 bar) using the Nose–Hoover thermostat algorithm and the Martyn–Tobias–Klein barostat algorithm, respectively. The resulting trajectories describing the structural and dynamic behavior of the protein–ligand complexes were then analyzed.

## 4. Conclusions

The QSAR model was developed in this research using the multiple linear regression statistical method. The model was expressed in terms of three descriptors, namely, polar surface area (PSA), dipole moment, and molecular weight, which showed a strong correlation with inhibitory activity and proved to be significant in anti-AChE activity. The developed QSAR model was statistically validated according to Golbarikh and Tropsha criteria, with obtained statistical parameters such as R^2^ = 0.701, R^2^test = 0.76, a Q^2^CV = 0.638, and RMSE = 0.336, as well as the application of a randomization test, VIF test, and applicability domain in order to demonstrate the reliability of the model and its satisfactory predictive ability for the anti-AChE activity of new compounds. The pharmacokinetic properties predicted by the ADMET test for the proposed compounds M1, M2, and M6 reveal that they have good ADMET properties and are non-toxic. They were then subjected to a molecular docking study, and the results revealed that the proposed compounds exhibited strong binding compared to Donepezil. Molecular dynamics simulation results also showed that compounds M1 and M2 have more stable interactions in the active site at 100 ns. The results of this study support the possibility that compounds M1 and M2 could be alternative inhibitors to Donepezil.

## Figures and Tables

**Figure 1 pharmaceuticals-17-00830-f001:**
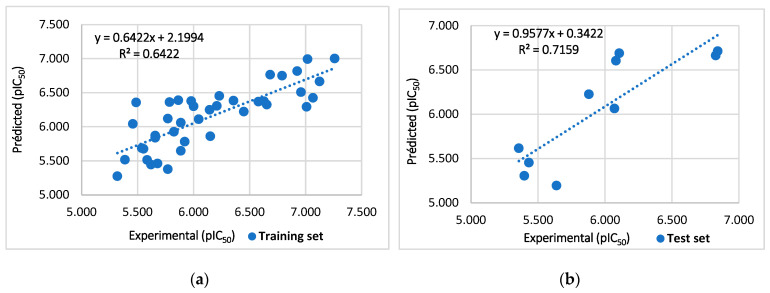
Graph of experimental pIC50 compared to overall predicted training (**a**) and the test set (**b**).

**Figure 2 pharmaceuticals-17-00830-f002:**
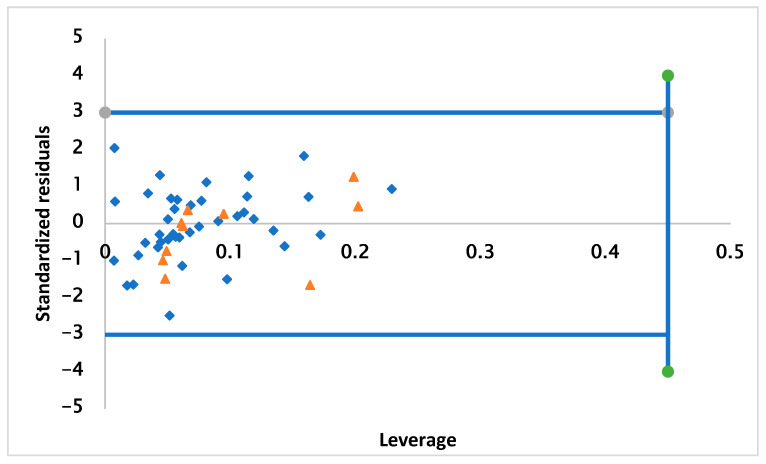
Graphical plot of QSAR model applicability domains.

**Figure 3 pharmaceuticals-17-00830-f003:**
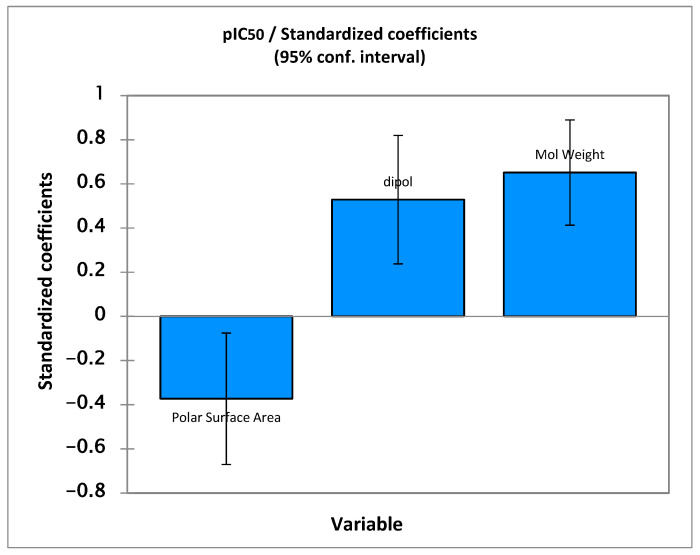
Characterization of the modeling by standardized coefficients.

**Figure 4 pharmaceuticals-17-00830-f004:**
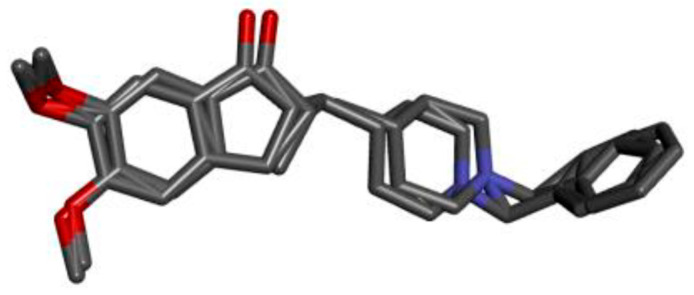
RMSD values and superposition of co-crystallized and docked compound in the active pocket of AChE.

**Figure 5 pharmaceuticals-17-00830-f005:**
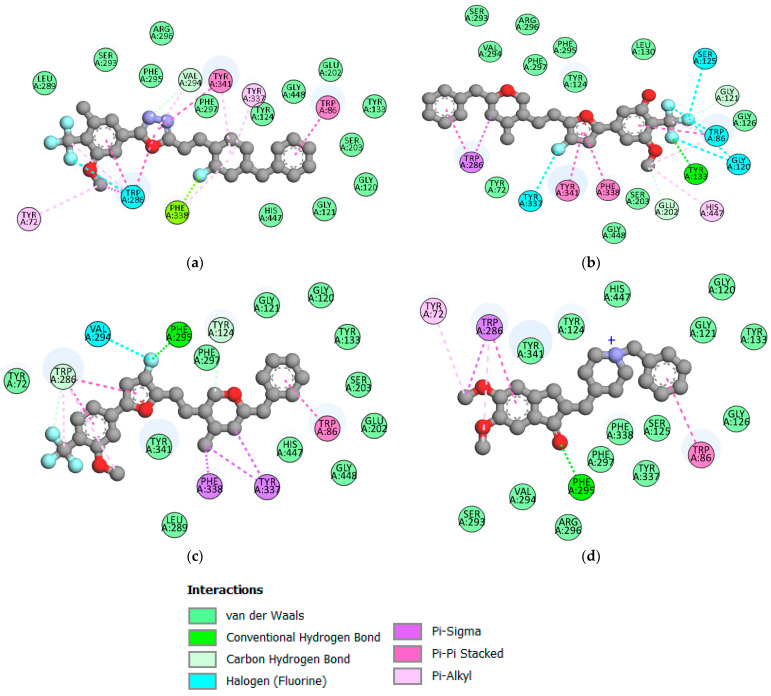
Two-dimensional interactions between AChE and the ligands M2 (**a**), M6 (**b**), M1 (**c**), and Donepezil (**d**).

**Figure 6 pharmaceuticals-17-00830-f006:**
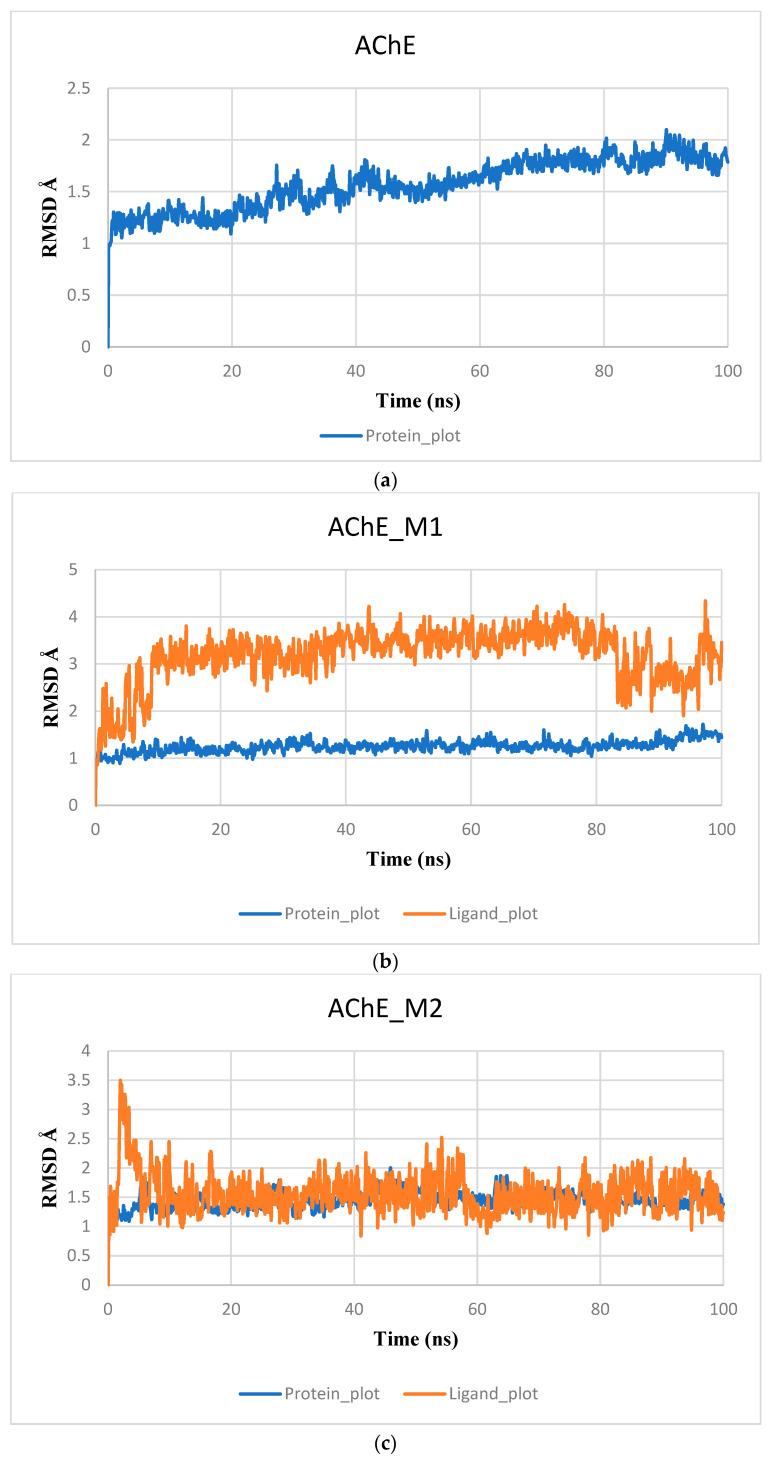
Protein–ligand root mean square deviation (RMSD) curves as a function of time of apo-protein AChE (**a**) and in a complex with the ligands M1 (**b**), M2 (**c**), and Donepezil (**d**).

**Figure 7 pharmaceuticals-17-00830-f007:**
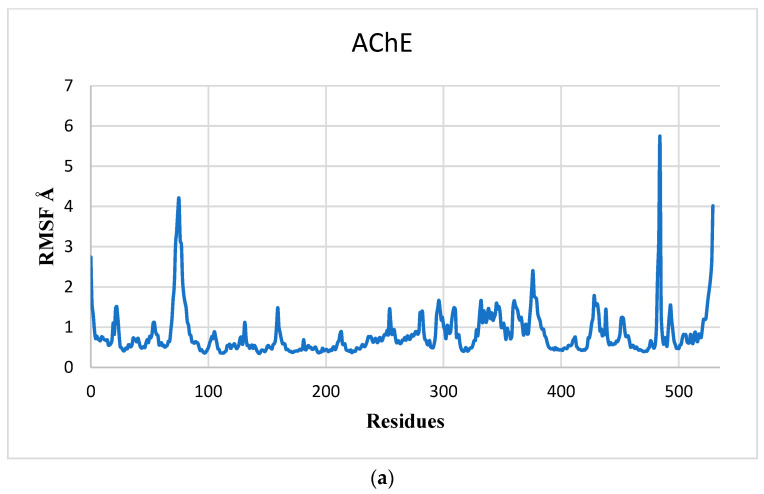
Time-dependent protein–ligand root mean square fluctuation (RMSF) curves AChE (**a**) in a complex with ligands M1 (**b**), M2 (**c**), and Donepezil (**d**).

**Figure 8 pharmaceuticals-17-00830-f008:**
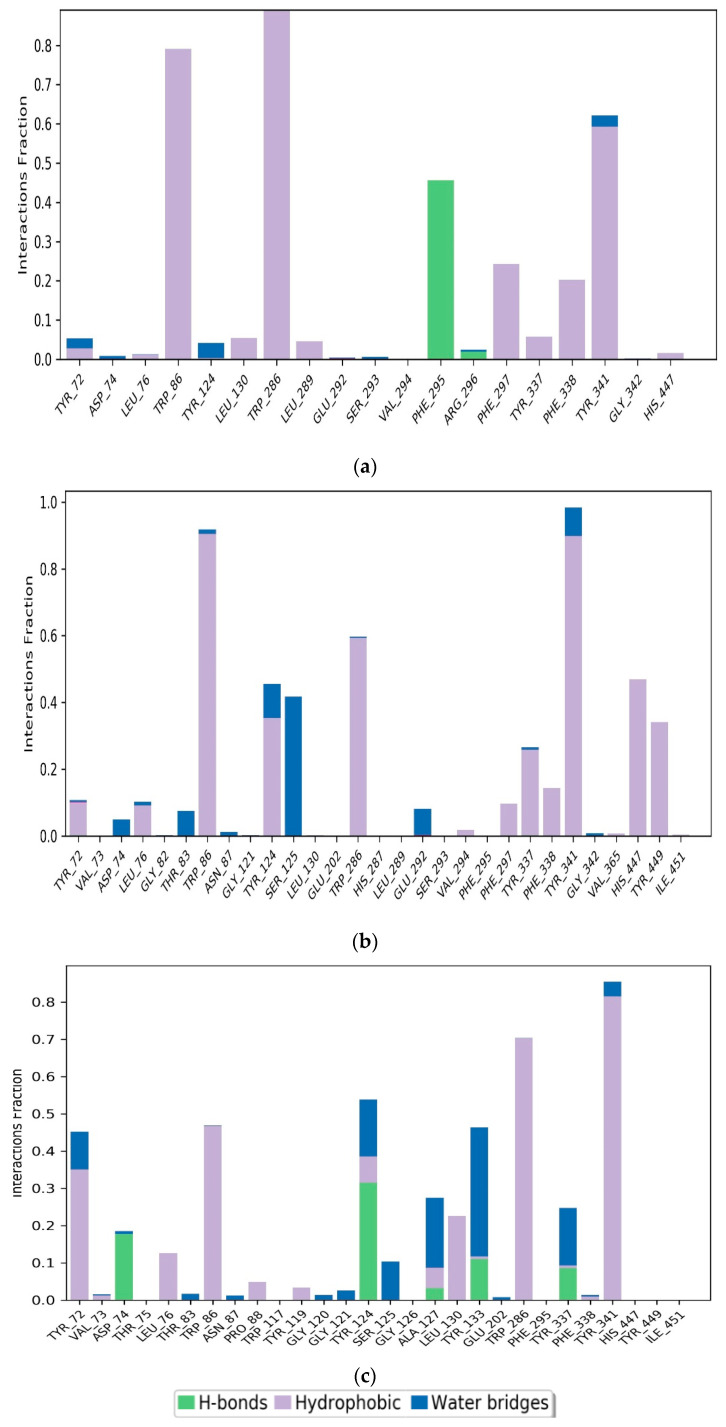
Simulation interaction plots and histograms of M2 (**a**), M1 (**b**), and Donepezil (**c**) in a complex with AChE.

**Table 1 pharmaceuticals-17-00830-t001:** Recommended minimum values for validation parameters for a generally accepted QSAR model [23].

Symbol Value	Name	Value
R^2^	Coefficient of determination	≥0.6
N_ext test set_	Minimum number of external test set	≥5
R^2^ − Q^2^_cv_	Difference between R^2^ and Q^2^_cv_	≤0.3
Q^2^_cv_	Cross-validation coefficient	>0.5
P _(95%)_	Confidence interval at 95% confidence level	˂0.05

**Table 2 pharmaceuticals-17-00830-t002:** Parameters of random models [24].

Parameters	Values	Threshold Value
Average r^2^	0.09	˂R^2^
Average Q^2^	−0.12	˂R^2^_CV_
CRp^2^	0.60	>0.500

**Table 3 pharmaceuticals-17-00830-t003:** Multi-collinearity statistics results.

Statistic	Polar Surface Area	Dipolar Moment	Mol Weight
VIF	2.16	2.069	1.388

**Table 4 pharmaceuticals-17-00830-t004:** Designed compounds.

Structures		Dipole Moment	PSA	Mol Wight	pIC_50_	hi	Outlier/ Inside
M1	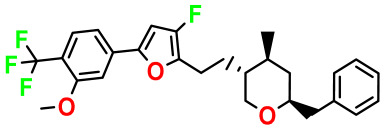	3.599	27.690	476.510	7.49	0.106	Inside
M2	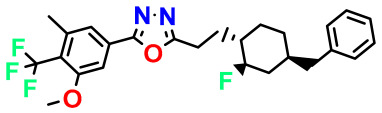	3.678	43.180	476.510	7.36	0.058	Inside
M3	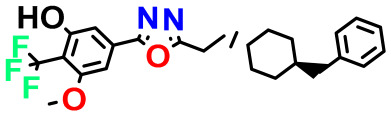	3.291	63.410	460.500	7.01	-	-
M4	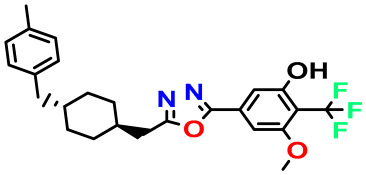	3.811	63.410	460.500	7.11	-	-
M5	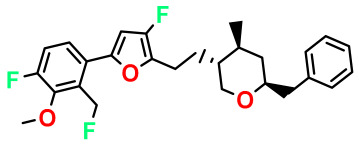	2.125	27.690	458.520	7.13	-	-
M6	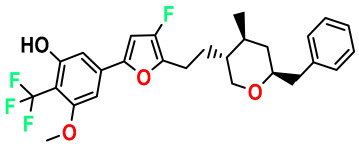	2.721	47.920	492.510	7.27	0.080	Inside
M7	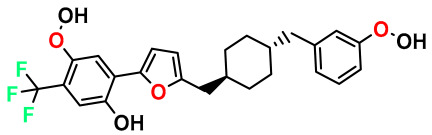	3.239	88.380	478.460	6.92	-	-
M8	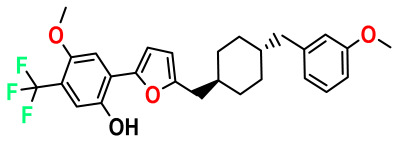	2.110	47.920	474.520	7.07	-	-
M9	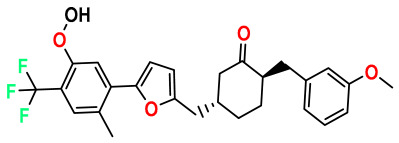	4.230	64.990	488.500	7.36	0.054	Inside
M10	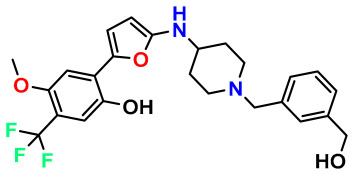	5.186	74.190	476.490	7.34	0.060	Inside
M11	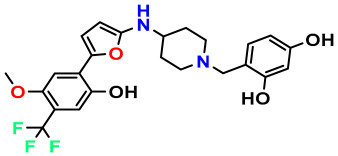	5.080	94.420	478.468	7.15	0.138	Inside

**Table 5 pharmaceuticals-17-00830-t005:** Drug similarity parameters according to Lipinski.

	logP	Molecular Weight	Num. H-Bond Acceptors	Num. H-Bond Donors
M1	6.86	476.50	7	0
M2	6.71	476.51	8	0
M6	6.47	492.50	8	1
M9	5.90	488.50	8	1
M10	4.40	476.49	8	3

**Table 6 pharmaceuticals-17-00830-t006:** ADMET properties of selected compounds.

	Absorption		Distribution		Metabolism	Excretion	Toxicity			
	CaCO_2_	HIA %	BBB Permeability (log BB)	CNS Permeability (log PS)	CYP2D6 Inhibitor	Total Clearance	AMES Toxicity	LD50	Hepatotoxicity	Skin Sensitization
M1	1.039	91.194	0.39	−1.292	No	0.336	No	2.553	No	No
M2	1.108	92.923	0.001	−1.371	No	0.261	No	2.752	No	No
M6	1.025	88.775	0.12	−1.56	No	0.214	No	2.366	No	No
M9	0.457	91.153	−0.96	−1.692	No	0.239	No	2.964	No	No
M10	0.947	88.76	−1.299	−2.193	No	0.647	No	2.85	Yes	No

**Table 7 pharmaceuticals-17-00830-t007:** Binding energies of compounds binding with 4EY7.

Ligands	M2	M1	M6	Donepezil
Score (kcal/mol)	−13	−12.6	−12.4	−10.8

**Table 8 pharmaceuticals-17-00830-t008:** Chemical structures and pIC50 values of 50 compounds.

pIC_50_ = 5.879 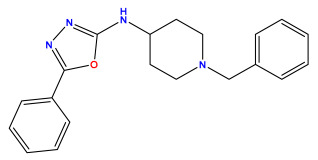	pIC_50_ = 6.106 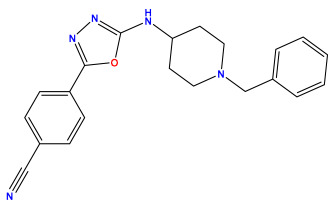	pIC_50_ = 6.827 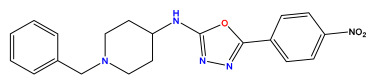
pIC_50_ = 6.578 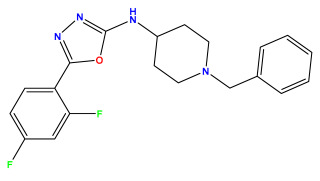	pIC_50_ = 6.082 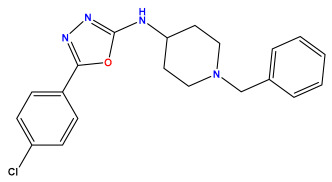	pIC_50_ = 7.125 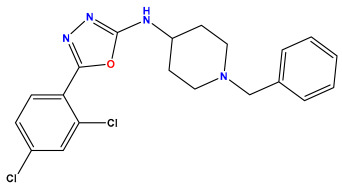
pIC_50_ = 7.260 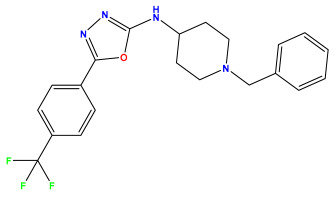	pIC_50_ = 7.018 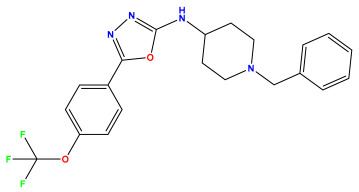	pIC_50_ = 5.770 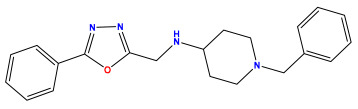
pIC_50_ = 5.979 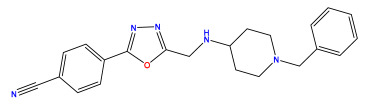	pIC_50_ = 6.654 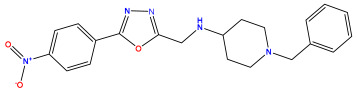	pIC_50_ = 6.449 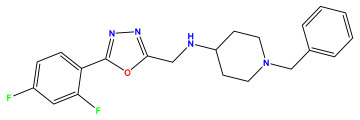
pIC_50_ = 5.785 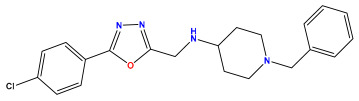	pIC_50_ = 7.066 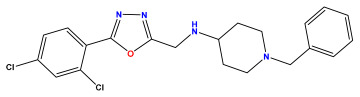	pIC_50_ = 6.842 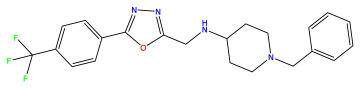
pIC_50_ = 6.924 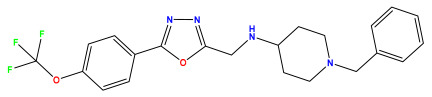	pIC_50_ = 5.863 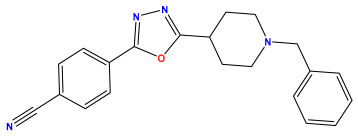	pIC_50_ = 6.629 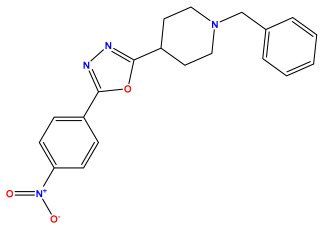
pIC_50_ = 5.457 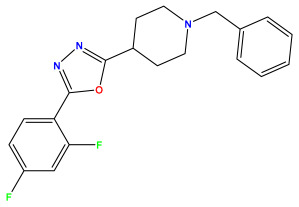	pIC_50_ = 7.009 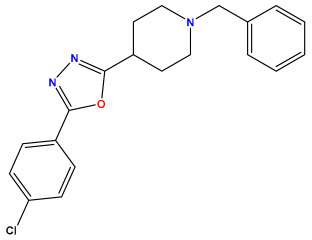	pIC_50_ = 5.487 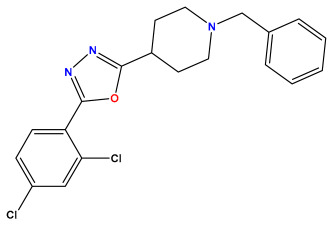
pIC_50_ = 6.684 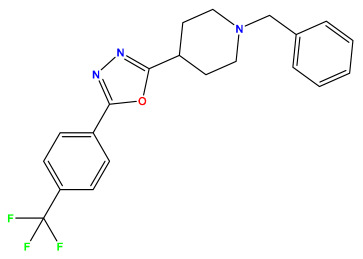	pIC_50_ = 6.790 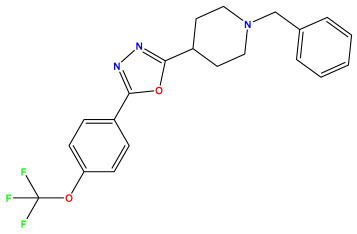	pIC_50_ = 5.678 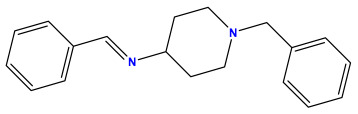
pIC_50_ = 5.553 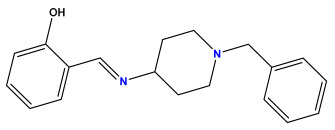	pIC_50_ = 5.319 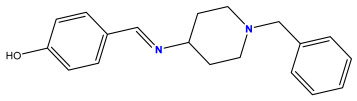	pIC_50_ = 5.585 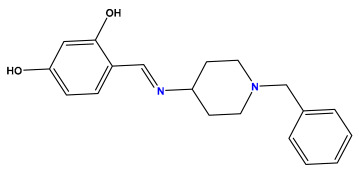
pIC_50_ = 5.357 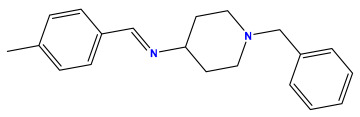	pIC_50_ = 6.046 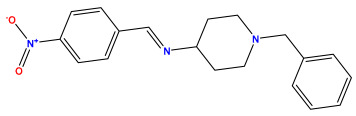	pIC_50_ = 6.357 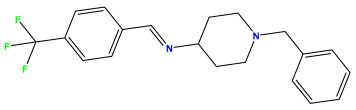
pIC_50_ = 6.000 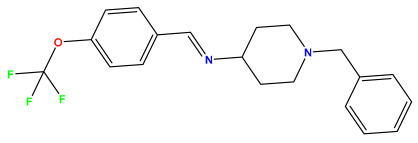	pIC_50_ = 5.824 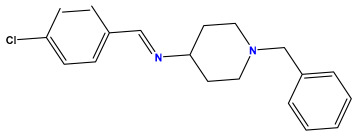	pIC_50_ = 5.886 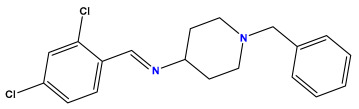
pIC_50_ = 5.538 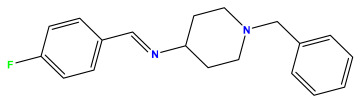	pIC_50_ = 5.886 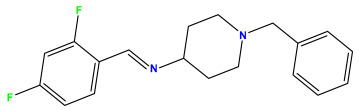	pIC_50_ = 5.658 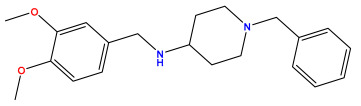
pIC_50_ = 5.770 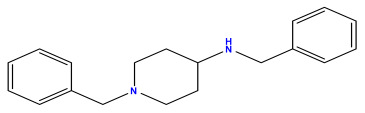	pIC_50_ = 5.658 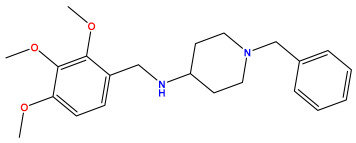	pIC_50_ = 5.620 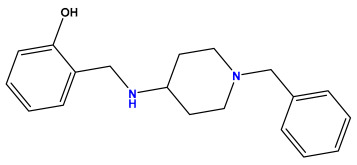
pIC_50_ = 5.398 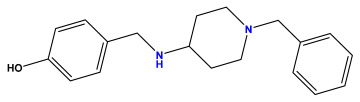	pIC_50_ = 5.638 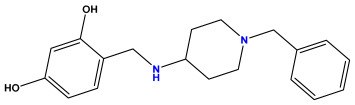	pIC_50_ = 5.387 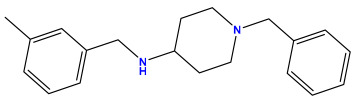
pIC_50_ = 5.432 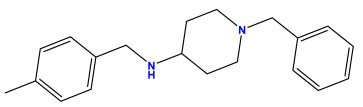	pIC_50_ = 6.143 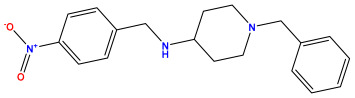	pIC_50_ = 6.959 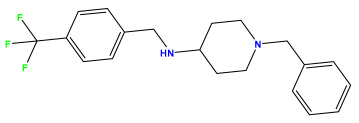
pIC_50_ = 6.230 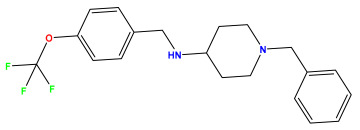	pIC_50_ = 6.071 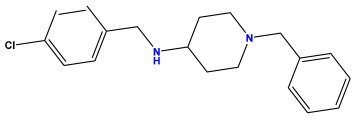	pIC_50_ = 6.208 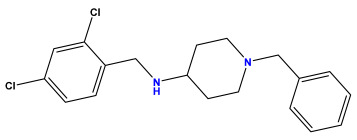
pIC_50_ = 5.921 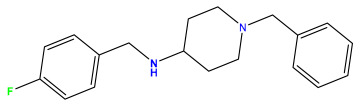	pIC_50_ = 6.149 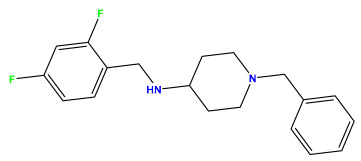	

## Data Availability

Data is contained within the article.

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
