# Peer review of "An In Silico Study Based on QSAR and Molecular Docking and Molecular Dynamics Simulation for the Discovery of Novel Potent Inhibitor against AChE"

_pharmaceuticals, 2024, doi:10.3390/ph17070830_

Round 1

Reviewer 1 Report

Comments and Suggestions for Authors

Review of the manuscript “An in-silico study based on QSAR, molecular docking and molecular dynamics simulation for the discovery of novel potent inhibitor against acetylcholinesterase”

The work is devoted to an in silico proposal of several structures which might act as potential inhibitors of AChE to act in turn as potential drugs for Alzheimer's disease. The motivation is well intended and supported with references from the literature.

Despite the compounds used to build QSAR model evidently span a narrow chemical space (at least narrower than that would be comfortable for the proposed structures), the overall study uses contemporary set of in silico tools usually applied for similar works. Certain points deserve attention of the authors (followed by correction) before the manuscript is published. I recommend to publish the manuscript after a major revision.

Specific points to address:

1. It should be pointed out and discussed somehow in the manuscript that the structures used for QSAR (Table 1) span a rather narrow chemical space of analogs of donepezil. In addition, it would be constructive to motivate the part of the molecule which is clearly identical for those structures in the training set – I mean the N-benzyl piperidine one.

2. In the docking study section it would be instructive to compare the binding modes or at least the aa residues involved for the designed compounds and the reference structure of donepezil (used in Fig.4 and the surrounding text without any reference to the compound) in Fig. 5.

3. In fig.4 donepezil looks somewhat strange. First, the amino group is not explicitly protonated, which may affect the docking results. Second, the piperidine moiety looks too planar, perhaps due to the projection used. I also suggest to visualize the experimental and docked structures with different color scheme to facilitate comprehension.

4. Although not reported in the current version of the manuscript a comparison of the MD simulation results of the proposed ligand with that of donepezil would be informative.

5. It is not quite clear from the methods section which of the two different versions of AutoDock was used within the study: either classical AutoDock4.2 or AutoDock Vina. Current version of AutoDock Vina (1.2.3) supports also calculating classical AutoDock4.2 scoring function in both docking and scoring modes. In case of AutoDock Vina careful charge assignment is not necessary, since the charges are not used internally.

Smaller” points:

1. When Alzheimer's disease’s abbreviation “(AD)” is first introduced in the manuscript it is correctly placed in the parentheses. But it surprises (not to say annoying) to see it multiple time after that in the text. So please do global replacement “(AD)” => “AD” except the first occurrence.

2. There are many sentences that are not complete, agreed and even contains the traces of initial versions. Please refine language and grammar.

Comments on the Quality of English Language

1. When Alzheimer's disease’s abbreviation “(AD)” is first introduced in the manuscript it is correctly placed in the parentheses. But it surprises (not to say annoying) to see it multiple time after that in the text. So please do global replacement “(AD)” => “AD” except the first occurrence.

2. There are many sentences that are not complete, agreed and even contains the traces of initial versions. Please refine language and grammar.

Author Response

  1. It should be pointed out and discussed somehow in the manuscript that the structures used for QSAR (Table 1) span a rather narrow chemical space of analogs of donepezil. In addition, it would be constructive to motivate the part of the molecule which is clearly identical for those structures in the training set – I mean the N-benzyl piperidine one.
  • We would like to thank you for your comments. We mentioned in the manuscript that the 2D QSAR model is constructed by a narrower set of molecules of analogues of N-benzylpiperidine and 5-phenyl-1,3,4-oxadiazoles.
  1. In the docking study section it would be instructive to compare the binding modes or at least the aa residues involved for the designed compounds and the reference structure of donepezil (used in Fig.4 and the surrounding text without any reference to the compound) in Fig. 5.
  • We would like to thank you for your comments. In the docking discussion section, we have added the residues involved in the designed compounds and the reference structure of donepezil, and we have also cited two references.
  1. In fig.4 donepezil looks somewhat strange. First, the amino group is not explicitly protonated, which may affect the docking results. Second, the piperidine moiety looks too planar, perhaps due to the projection used. I also suggest to visualize the experimental and docked structures with different color scheme to facilitate comprehension.
  • We would like to thank you for your pertinent observations concerning the representation of donepezil in Figure 4. We have taken your comments into account and have made the necessary adjustments. Firstly, we have explicitly protonated the amino group of donepezil, which has influenced the docking results. These changes have been incorporated into the manuscript. Secondly, we understand your concerns about the piperidine moiety appearing too flat in the 2D visualization. We would like to point out that this appearance can be attributed to the two-dimensional representation. For the suggestion to visualize the experimental and anchored structures with different colors. We have taken this recommendation into consideration
  1. Although not reported in the current version of the manuscript a comparison of the MD simulation results of the proposed ligand with that of donepezil would be informative.
  • We thank you for your pertinent suggestion. We have taken this comment into account and have included in the manuscript a comparison of MD simulation results for the proposed ligand with those for donepezil.
  1. It is not quite clear from the methods section which of the two different versions of AutoDock was used within the study: either classical AutoDock4.2 or AutoDock Vina. Current version of AutoDock Vina (1.2.3) supports also calculating classical AutoDock4.2 scoring function in both docking and scoring modes. In case of AutoDock Vina careful charge assignment is not necessary, since the charges are not used internally.
  • In response to your comment regarding the specific version of AutoDock used in our study, we acknowledge the omission and wish to provide the following clarification:
  • In our study, we used AutoDock Vina (version 1.5.7) for docking simulations. This version was chosen because of its improved performance and its support for the AutoDock 4.2 scoring function.
  • Regarding loads, we confirm that AutoDock Vina does not require detailed loads as it does not use them in its internal calculations. Nevertheless, we have followed standard protocols for the preparation of our ligands and receptors to ensure the consistency and accuracy of our simulations

“Smaller” points:

  1. When Alzheimer's disease’s abbreviation “(AD)” is first introduced in the manuscript it is correctly placed in the parentheses. But it surprises (not to say annoying) to see it multiple time after that in the text. So please do global replacement “(AD)” => “AD” except the first occurrence.
  • Thank you for your comments. We have taken your comments into account and made the corrections in the manuscript.
  1. There are many sentences that are not complete, agreed and even contains the traces of initial versions. Please refine language and grammar.
  • Thank you for bringing these points to our attention. We are working hard to improve the quality of our work.

Reviewer 2 Report

Comments and Suggestions for Authors

In the manuscript (ID: pharmaceuticals-3028832), the authors in the study designed new potential inhibitors of AChE using the QSAR tool, in addition to resorting to the use of other modeling techniques such as molecular docking and molecular dynamics.

The manuscript requires correction.

1) Which of compounds belong to the test set and to the training set?

2) There are no information how the compounds were designed and if there are any experimental studies on their binding affinity to AChE.

3) The table 5 should be corrected such as structures of compounds should be more clear shown.

4)  Why in Tables 6 and 7 are only selected molecules shown, there is no explanation,

5) The table 7 is in the text as table 8.

6) The sentences e.g. In Figure 5 (a), the compound M1 docks with AChE having established in their contact van der Wals inter- actions by these residues Tyr133, Ser203, Gly120, Gly121, His447 and Pi-Pi Stacked type interactions implied by two residues Trp86, Trp286, Tyr341 and pi-alkyl type interactions mediated by Tyr72, Trp286, Val294, Tyr337, Phe338, Tyr341 and pi-sigma binding have been identified between the M1 ligand and AChE with the residue of Trp286, thus the formation of a hydrogen bond by Val294, and the formation of a halogen bond by Trp286, and a Pi-Lone Pair type bond formed by Phe338” are too long and should be corrected. The other sentences in this part of the manuscript should be also corrected.

7) In addition, the authors write “implied by two residues Trp86, Trp286, Tyr341” but there are three residues. Should be corrected.

8) The Figure 6 should be divided in two figures where MD for M1 and M2 will be presented separately from MD of AChE and complexes.

9) The quality of Figure 8 should be improved.

10) Which residues in AChE are important from mutagenesis studies for ligand binding and provide this information with discussion in the text.

Author Response

1) Which of compounds belong to the test set and to the training set?

  • Thank you for your question. We have taken your suggestion into account and have added a legend in our manuscript to clearly distinguish the compounds belonging to the learning set from those belonging to the test set. Figure 1

2) There are no information how the compounds were designed and if there are any experimental studies on their binding affinity to AChE.

  • Thank you for your comments. We have taken into account your observation that there is no information on the design of the compounds in the manuscript. We would like to point out that details on the design of the proposed compounds have already been mentioned in the text. However, we acknowledge that there is no information on experimental studies regarding their binding affinity to AChE.

3) The table 5 should be corrected such as structures of compounds should be more clear shown.

  • Thank you for your comment on Table 5 (Table 4 in revised version). We have taken your suggestion into consideration and have made the necessary corrections to improve the clarity of the compound structures.

4)  Why in Tables 6 and 7 are only selected molecules shown, there is no explanation,

  • Thank you for your comment. We have already indicated in the manuscript that the molecules presented in Tables 6 and 7 were selected because of their higher activity than that of the model molecule and their inclusion in the applicability domain of the model. On the other hand, the other designed molecules were rejected because of their lower activity than that of the model molecule and their position outside the applicability domain of the model. It is important to note that the molecules M3, M4, M5, M7 and M8 are not examined in the ADMET study, as they were rejected.

5) The table 7 is in the text as table 8.

  • Thank you for your comment. We have corrected updated the tables numbering

6) The sentences e.g. In Figure 5 (a), the compound M1 docks with AChE having established in their contact van der Wals inter- actions by these residues Tyr133, Ser203, Gly120, Gly121, His447 and Pi-Pi Stacked type interactions implied by two residues Trp86, Trp286, Tyr341 and pi-alkyl type interactions mediated by Tyr72, Trp286, Val294, Tyr337, Phe338, Tyr341 and pi-sigma binding have been identified between the M1 ligand and AChE with the residue of Trp286, thus the formation of a hydrogen bond by Val294, and the formation of a halogen bond by Trp286, and a Pi-Lone Pair type bond formed by Phe338” are too long and should be corrected. The other sentences in this part of the manuscript should be also corrected.

  • Thank you for your comment. We have corrected this paragraph

7) In addition, the authors write “implied by two residues Trp86, Trp286, Tyr341” but there are three residues. Should be corrected.

  • Thank you for your comment. We have corrected the error.

8) The Figure 6 should be divided in two figures where MD for M1 and M2 will be presented separately from MD of AChE and complexes.

  • Thank you for your comment. We have taken this into consideration.

9) The quality of Figure 8 should be improved.

  • Thank you for your comment. We have taken this into consideration

10) Which residues in AChE are important from mutagenesis studies for ligand binding and provide this information with discussion in the text.

  • Thank you for your suggestion. We have taken your request into account and have added information on the residues important in AChE for mutagenesis studies concerning ligand binding in the manuscript.

Reviewer 3 Report

Comments and Suggestions for Authors

Khedraoui et al., present a virtual screening using several methods, but mostly developed from QSAR modelling. The methods are correctly conducted; however, I have the following concerns:

While QSAR modelling follows good practices, there are some missing aspects that could and should be adressed. For instance, there is some comment to the descriptors used for model construction. Yet, this is very brief and general. I do think that in this case the fifth OCDE principle for QSAR modelling can be applied: "A mechanistic interpretation if possible"

Now, as for the structure-based methods, there is no comparison with a reference compound. Therefore, it is difficult to determine the significance nor value of the presented results. My point is further supported based on the molecular interactions from molecular dynamics. For M1 & M2, hydrophobic interaction are dominant during the simulation. This could be related to the role of PSA, yet as previously mentioned it is the authors' task to develop and establish these.

Finally, while the results do seem promising, it is difficult to get a baseline on the putative affinity of the proposed ligands. More so, taking the fact that even 100ns may not be enough to determine this. It often happens that works argue on stability. Based solely on RMSD or RMSF values, which in turn are highly variable, between MD runs. Hence, a more robust approach is necessary.

Comments on the Quality of English Language

There are several grammar mistakes in the main text. In some cases there are confusing sentences or even incomplete ideas; e.g., [...] deficiency of this neurotransmitter ACh leads to cognitive disorders [16], On the Based on these facts the therapeutic approach currently considered [...]

Similarly, there are many long sentences.

Author Response

Khedraoui et al., present a virtual screening using several methods, but mostly developed from QSAR modelling. The methods are correctly conducted; however, I have the following concerns:

While QSAR modelling follows good practices, there are some missing aspects that could and should be adressed. For instance, there is some comment to the descriptors used for model construction. Yet, this is very brief and general. I do think that in this case the fifth OCDE principle for QSAR modelling can be applied: "A mechanistic interpretation if possible"

  • We thank you for your concerns. We have taken your suggestion into account and have added a mechanistic interpretation of the descriptors used to build the QSAR model in the manuscript.

Now, as for the structure-based methods, there is no comparison with a reference compound. Therefore, it is difficult to determine the significance nor value of the presented results. My point is further supported based on the molecular interactions from molecular dynamics. For M1 & M2, hydrophobic interaction are dominant during the simulation. This could be related to the role of PSA, yet as previously mentioned it is the authors' task to develop and establish these.

  • Thank you for your feedback. We have taken into consideration your observation regarding the lack of a reference compound comparison in our study. As a result, we have included a comparison with a reference compound in our analysis to provide a basis for assessing the significance and value of the results presented.

Finally, while the results do seem promising, it is difficult to get a baseline on the putative affinity of the proposed ligands. More so, taking the fact that even 100ns may not be enough to determine this. It often happens that works argue on stability. Based solely on RMSD or RMSF values, which in turn are highly variable, between MD runs. Hence, a more robust approach is necessary.

  • Thank you for your comments and suggestions. We have added an MM/GBSA study to provide a more robust assessment of the affinity of the proposed ligands. This strengthens the quality of our study.

There are several grammar mistakes in the main text. In some cases there are confusing sentences or even incomplete ideas; e.g., [...] deficiency of this neurotransmitter ACh leads to cognitive disorders [16], On the Based on these facts the therapeutic approach currently considered [...]

Similarly, there are many long sentences.

  • Thank you for bringing these points to our attention. We are working hard to improve the quality of our work.

Round 2

Reviewer 1 Report

Comments and Suggestions for Authors

The authors have addressed the points I made in the first review, so I recommend to publish the manuscript.

Author Response

Dear reviewer,

On behalf of our research team, we would like to express our sincere gratitude to appreciate the time and effort that you have devoted to reviewing our manuscript. 

Reviewer 2 Report

Comments and Suggestions for Authors

The manuscript  was corrected according to review suggestions. The compound structures in the Table 4 should be improved. Some sentences are too long e.g. lines 287-290.  In the text between lines 335-355 the number of Figure described is missing. The spelling of amino acids should be the same throughout the text and meanwhile it is in capital letters or as a proper name.

Author Response

  • Thank you for your revision suggestions. We have taken your feedback into account and addressed the mentioned points.
  • The structures of the compounds in Table 4 have been improved, overly long sentences have been shortened, the missing figure number has been added in the text between lines 335-355, and we have ensured consistency in the spelling of amino acids throughout the text.

Reviewer 3 Report

Comments and Suggestions for Authors

I thank the authors for taking my comments as constructive criticism of their work. While I do acknowledge the improvement and effort to address my observations. I must say that there are persisting issues in the manuscript:

QSAR modelling is methodologically correct, yet I ponder on how did the authors yielded three common descriptors. The relation between any of these and biological activity can be very vague. Thus, my comment on mechanistic interpretation. Which was indeed futile as the authors attempt at it was a very redundant text with no contribution whatsoever.

While on this subject, this is an example of a section lacking important details. As it is stated: "the electronic descriptors were calculated using the Gauss View program [48], adopting the approach of the hybrid density functional theory B3LYP combining the three Becke parameters and the Lee-Yang- Parr exchange correlation functional with basis groups 6-31G (d, p), and by the Chem3d software we calculated different molecular descriptors, in particular constitutional descriptors, physicochemical descriptors, geometric descriptors [47] which will be used as input variables in the establishment of a 2D QSAR model, because they quantitatively translate the information chemicals relative to molecular structures [49]."

While the paragraph is descriptive I ponder on why did the number of calculated descriptors is on a different section? Also, I must say that having 50 descriptors is an interesting contrast to other studies where more than a hundred are considered. This is not to say that more descriptors shall be better or the norm. As I said I ponder as to why overlook more well-known programms such as DataWarrior or even RDKit.

Similarly, the authors did carry out endpoint methods. But any description on their methodology is non-existent. This is serious as results are highly dependent on parameter selection and overall implementation. Plus, this helps me to introduce another question. When compared, donepezil engages in several hydrogen bonds. Then why is that hit compounds exhibit higher affinity when no such interaction is present. Either in number, persistence or with the same residues as donepezil?

I stand with my initial appreciation, the work is methodologicaly correct but I fail to grasp any significant contribution from the QSAR modelling and the proposed hits by it.

Comments on the Quality of English Language

Some style erros persist, for example spelling donepezil as done-pezil. In addition, sentences with confusing wording are still present. Take for instance, the beginning sentence of section 3.2.

Author Response

Dear Reviewer,

Thank you for your feedback and for taking the time to review our work in detail. We sincerely appreciate your constructive comments and the effort you have put into helping us improve our manuscript.

We acknowledge the persistent issues you have mentioned and we have addressed them in this revised version. Your expertise and suggestions are invaluable to us and help us advance our research.

Best regards

QSAR modelling is methodologically correct, yet I ponder on how did the authors yielded three common descriptors. The relation between any of these and biological activity can be very vague. Thus, my comment on mechanistic interpretation. Which was indeed futile as the authors attempt at it was a very redundant text with no contribution whatsoever.

  • Thank you for your comment. We have noted your concern regarding the three common descriptors and their relation to biological activity. Our QSAR model has demonstrated a significant correlation between the descriptors and pIC50. Here are more detailed explanations on the relevance of each descriptor:
  1. Molecular Weight (MW): Larger molecules have a greater contact surface, possess more atoms and functional groups, allowing for more numerous and stronger non-covalent interactions with the binding sites of AChE (such as hydrogen bonds, hydrophobic interactions, and van der Waals forces), which can enhance the molecule's affinity for the target enzyme, and thus the inhibitory activity.
  2. Dipole Moment (µ): The dipole moment of a molecule is a measure of the distribution of electrical charges. A molecule with a high dipole moment has a net charge, which can favor electrostatic interactions with the charged residues of the target enzyme, thereby strengthening the binding between the ligand and the protein.
    • In our regression model, the dipole moment is positively correlated with anti-AChE activity, indicating that increasing the molecule's polarity by substituting polar groups will result in increased anti-AChE activity.
  3. Polar Surface Area (PSA): PSA (polar surface area) represents the molecular polar surface area and is defined as the area due to nitrogen and oxygen and any attached hydrogen, i.e., the molecular surface sum, usually van der Waals, over all polar atoms.
    • A high PSA can decrease the molecule's permeability through cell membranes, limiting accessibility to the enzymatic target. Conversely, a reduced PSA can improve this accessibility, thereby increasing biological activity.
    • In the regression model, PSA is negatively correlated with anti-AChE activity, indicating that nitrogen and oxygen atoms have a negative influence on anti-AChE activity. Consequently, the increase in the pIC50 value of the designed compounds is due to the absence of additional nitrogen and oxygen atoms.

We hope this explanation clarifies our approach and justifies the relevance of the selected descriptors as well as the importance of their mechanistic interpretation. We remain available for any further questions or clarifications.

While on this subject, this is an example of a section lacking important details. As it is stated: "the electronic descriptors were calculated using the Gauss View program [48], adopting the approach of the hybrid density functional theory B3LYP combining the three Becke parameters and the Lee-Yang- Parr exchange correlation functional with basis groups 6-31G (d, p), and by the Chem3d software we calculated different molecular descriptors, in particular constitutional descriptors, physicochemical descriptors, geometric descriptors [47] which will be used as input variables in the establishment of a 2D QSAR model, because they quantitatively translate the information chemicals relative to molecular structures [49]."

While the paragraph is descriptive I ponder on why did the number of calculated descriptors is on a different section? Also, I must say that having 50 descriptors is an interesting contrast to other studies where more than a hundred are considered. This is not to say that more descriptors shall be better or the norm. As I said I ponder as to why overlook more well-known programms such as DataWarrior or even RDKit.

  • Thank you for your insightful comment. Regarding your first question about the distribution of sections, we chose to mention in the article that we calculated 50 descriptors without displaying the specific descriptors. This approach aims to maintain the conciseness and readability of the article while providing the necessary information about the total number of descriptors used in the analysis.
  • As for the choice of software and descriptors, we obtained a set of 50 descriptors after performing a preliminary analysis and dimension reduction based on the correlation matrix. Although other studies often use a larger number of descriptors, our approach was to focus on those that show a strong correlation with the biological activity pIC50. The goal was to simplify the model while maintaining its robustness and biological relevance.
  • Regarding the use of programs like DataWarrior or RDKit, we chose Gauss View and Chem3D for this specific study due to their easy integration with our computational workflow and their ability to provide the necessary descriptors for our model.

Similarly, the authors did carry out endpoint methods. But any description on their methodology is non-existent. This is serious as results are highly dependent on parameter selection and overall implementation. Plus, this helps me to introduce another question. When compared, donepezil engages in several hydrogen bonds. Then why is that hit compounds exhibit higher affinity when no such interaction is present. Either in number, persistence or with the same residues as donepezil?

  • Thank you for your pertinent comments. To answer your question regarding the interactions of the compounds with AChE in comparison to donepezil, our results show that donepezil forms only one hydrogen bond with AChE, which is consistent with previous studies. In contrast, our tested compounds exhibit higher affinity by forming a sufficient number of other types of bonds and interactions. This justifies their higher scores compared to donepezil.
  • Our objective is to propose molecules with better affinity for AChE than donepezil, and the results confirm that our compounds can potentially surpass donepezil in terms of interactions and overall affinity.

I stand with my initial appreciation, the work is methodologicaly correct but I fail to grasp any significant contribution from the QSAR modelling and the proposed hits by it.

  • Thank you for your comment. We maintain that the methodology used is correct and would like to emphasize the significant contribution of our work. The QSAR modeling we employed has enabled us to propose new molecules with higher anti-AChE activity than existing ones. This advancement is important for the development of more effective AChE inhibitors.

Comments on the Quality of English Language

Some style erros persist, for example spelling donepezil as done-pezil. In addition, sentences with confusing wording are still present. Take for instance, the beginning sentence of section 3.2.

  • Thank you for your detailed feedback. We have corrected the style errors, notably the spelling of "donépézil." Additionally, we have reviewed the confusing formulations. For example, the first sentence of section 3.2 has been rephrased to be clearer and more concise.

Round 3

Reviewer 3 Report

Comments and Suggestions for Authors

No further comment

Comments on the Quality of English Language

Make an additional revision of the text